# Determinants of Pre-Surgical Treatment in Primary Rectal Cancer: A Population-Based Study

**DOI:** 10.3390/cancers15041154

**Published:** 2023-02-10

**Authors:** Israa Imam, Klara Hammarström, Bengt Glimelius

**Affiliations:** Department of Immunology, Genetics and Pathology, Uppsala University, SE-751 85 Uppsala, Sweden

**Keywords:** rectal cancer, radiotherapy, chemoradiotherapy, preoperative therapy, neoadjuvant therapy, magnetic resonance imaging, treatment guidelines, population-based

## Abstract

**Simple Summary:**

Preoperative radiotherapy has an established role in the treatment of rectal cancer, alone or with chemotherapy, but the use varies considerably. Many scientists have strived to reduce the use of radiation while maintaining high local control rates, partly counterbalanced by an ambition to preserve the organ. Besides patient-related factors, stage as defined by magnetic resonance imaging (MRI) is most important for the decision at multidisciplinary team (MDT) conferences to recommend direct surgery or any treatment prior to (eventual) surgery. In a large prospective, unselected and properly staged patient cohort, MRI characteristics were most important for treatment selection, but patient-related factors were also relevant. Changes over time, reflecting changed national guidelines that were striving to reduce the use of radiation, were seen; however, they were probably interpreted differently in the two analysed regions. The accuracy of MRI evaluated by specially trained radiologists, during an MDT conference in real life, was poor.

**Abstract:**

When preoperative radiotherapy (RT) is best used in rectal cancer is subject to discussions and guidelines differ. To understand the selection mechanisms, we analysed treatment decisions in all patients diagnosed between 2010–2020 in two Swedish regions (Uppsala with a RT department and Dalarna without). Information on staging and treatment (direct surgery, short-course RT, or combinations of RT/chemotherapy) in the Swedish Colorectal Cancer Registry were used. Staging magnetic resonance imaging (MRI) permitted a division into risk groups, according to national guidelines. Logistic regression explored associations between baseline characteristics and treatment, while Cohen’s kappa tested congruence between clinical and pathologic stages. A total of 1150 patients without synchronous metastases were analysed. Patients from Dalarna were older, had less advanced tumours and were pre-treated less often (52% vs. 63%, *p* < 0.001). All MRI characteristics (T-/N-stage, MRF, EMVI) and tumour levels were important for treatment choice. Age affected if chemotherapy was added. The correlation between clinical and pathological T-stage was fair/moderate and poor for N-stage. The MRI-based risk grouping influenced treatment choice the most. Since the risk grouping was modified to diminish the pre-treated proportion, fewer patients were irradiated with time. MRI staging is far from optimal. A stronger wish to decrease irradiation may explain why fewer patients from Dalarna were irradiated, but inequality in health care cannot be ruled out.

## 1. Introduction

Preoperative radiotherapy has an established role in the treatment of rectal cancer, alone (RT) or with chemotherapy (CRT) [1]. It is primarily used in locally advanced “non-resectable” cancers to allow radical surgery after tumour downsizing/downstaging, and in less advanced, resectable tumours to lower local recurrence rates [2,3,4].

Much of the knowledge behind the treatment recommendations in rectal cancer relies on randomized trials [2,5,6,7,8,9,10,11]. The well-being of the patients has also been investigated [12,13,14], allowing an evaluation of the balance between gains and losses due to the adverse effects after RT/CRT [15,16]. The recommendations and use vary considerably [17,18,19,20], reflecting different values put on the radiation-induced negative effects [21]. Many scientists have strived to reduce the use of radiation while maintaining high local control rates [22,23,24]. This has partly been counterbalanced by an ambition to preserve the organ; tumours exempted from pre-treatment are the early, often small tumours with low local recurrence risks, the ones with the greatest chance to respond completely [25,26,27,28].

Besides patient-related factors, the stage defined by magnetic resonance imaging (MRI) is important for the decision at multidisciplinary team (MDT) conferences to recommend direct surgery or any treatment prior to (eventual) surgery. MRI is technically developing and the interpretation of the images is subject to difficulties, resulting in variabilities in the evaluations [29,30]. Of particular concern is the evaluation of the nodal stage [31].

We have used a large prospective collection of information from an unselected, properly staged patient cohort, to understand selection mechanisms to different treatments and changes over time. Two adjacent Swedish regions were compared for the evaluation of equal care, being an important aspect of Swedish health care. Given the importance of MRI for tumour staging, we also aimed to explore the accuracy of MRI evaluated by specially trained radiologists during an MDT conference in real life.

## 2. Materials and Methods

### 2.1. Patients

All patients living in two adjacent regions in Sweden (Uppsala: population 388,394 in 2020; Dalarna: population 287,676 in 2020) and diagnosed between 2010 and 2020 were identified from the Swedish Colorectal Cancer Registry (SCRCR) [32] and a biobank initiative (U-CAN) initiated in June 2010 [33]. Clinical information containing staging, treatment and follow-up was obtained from the SCRCR. Missing information was collected from clinical records, including the MDT reports, and from the re-evaluation of images and pathological slides, as described [27].

### 2.2. Staging and Treatment

Staging included a clinical work-up, computed tomography of the thorax, abdomen and pelvis, and MRI of the pelvis [34]. Every case was discussed at an MDT conference held in Uppsala or Falun, Dalarna with the participation of surgeons, GI clinical oncologists (all belonged to the same team in Uppsala), specially trained radiologists [35,36], pathologists and nurses. Since Dalarna does not have radiation equipment, patients from Dalarna to be irradiated were discussed also in Uppsala; in the case of a discrepancy, the MDT decision in Uppsala was used.

MRI parameters of relevance were clinical (c)T-stage with subdivisions of cT3 and cT4, distance to the mesorectal fascia (MRF), extramural vascular invasion (EMVI) and lymph node engagement [37]. Based upon tumour level, measured with a rigid rectoscope, and MRI stage, a risk grouping was made according to the “good–bad–ugly” concept [38] (Appendix A).

National guidelines for colorectal cancer were published in 2008, 2016 and 2020 [39,40] but used routinely the year before being published. Early/good tumours were operated on directly with an abdominal procedure or, if polypoid (chiefly cT1N0), with a local procedure; intermediate/bad tumours received short-course RT (scRT; 5x5 Gy in one week) with direct or delayed surgery according to the Stockholm III trial [11]; and locally advanced/ugly tumours received CRT (50 Gy in 5 weeks with a fluoropyrimidine, 5FU/leucovorin or capecitabine, Appendix A). Between 2010 and 2020, ongoing trials (Stockholm III, RAPIDO [41], LARCT-US [42]) could modify which treatment the patients received (Appendix A). An alternative to CRT according to the RAPIDO/LARCT-US protocols was scRT + chemotherapy (scRT + CT; 5x5 Gy followed by 3–5 months of chemotherapy before surgery). The reference treatment for patients participating in a randomized trial was used.

### 2.3. Statistics

The continuous variable age was analysed using Mann–Whitney U test for categorical variables, differences between groups were investigated using the Chi-square test of independence. Logistic regression analyses were performed of associations between patient characteristics, the two regions, year of diagnosis, tumour level and MRI characteristics defining risk group and selection to preoperative therapy (RT or CRT/scRT + CT). Univariable logistic regression was used to identify predictors of receiving preoperative treatment. Variables likely to be associated with receiving preoperative treatment were included in a multivariable logistic regression based on a statistical significance of *p* < 0.1. Adjustments were made for tumour level, region and year of diagnosis. Age and gender were also included. The level of statistical significance was set at *p* < 0.05. Cohen’s kappa was used to evaluate the correlation between clinical and pathological T- and N-stage, and considered poor (<0.20), fair (0.21–0.40), moderate (0.41–0.60), good (0.61–0.80), or excellent (>0.80). Statistical analyses were performed using IBM^®^SPSS^®^ statistical software version 27.0 and R4.1.3 (R Foundation for Statistical Computing).

## 3. Results

### 3.1. Patient Characteristics

Between 2010–2020, 1481 rectal cancer patients (700 in Uppsala and 781 in Dalarna) were diagnosed (Figure 1), corresponding to annual incidences of 16.4 and 24.7/100,000, respectively. Virtually all patients were staged as recommended. A pelvic MRI was performed on 91% of the patients with no difference between regions. Reasons for no MRI were due to the presence of a pacemaker or an endoscopically removed polypus tumour. Patient characteristics are described in Table 1 (non-metastatic, M0) and Appendix A (all patients). A total of 23 and 22%, respectively, had distant metastases (M1) at diagnosis in Uppsala and Dalarna; M1 patients had markedly more advanced primary tumours than M0 patients. No major differences in characteristics were seen in M0 patients according to sex, cT-stage, and tumour level between the regions, but patients from Dalarna were older (*p* < 0.001).

### 3.2. Changes in cT- and cN-Stages and Other MRI Characteristics with Time

cT-stage or the presence of EMVI did not change over time, but EMVI positivity increased with increasing cT and cN-stages (Appendix A). More cN+ stages were diagnosed in Uppsala than in Dalarna (62% vs. 52%, *p* = 0.003). In both regions, after an initial increase from 2010 to 2013 (Appendix A), this proportion decreased over time (*p* < 0.001). In particular, cN2 was seen less frequently with time (*p* < 0.001, Figure 2a).

### 3.3. Risk Grouping According to the “Good–Bad–Ugly” Concept

Significantly more tumours were locally advanced/ugly in Uppsala than in Dalarna (39% vs. 31%, *p* = 0.01) as a consequence of more tumours being considered as cN+, MRF+ and EMVI+. The proportions considered intermediate/bad were approximately the same and, consequently, more tumours were classified as early/good in Dalarna (Table 1).

Changes in the guidelines (see Appendix A) over the years meant that more tumours were classified as early/good or intermediate/bad during the two recent periods. Re-classification of all tumours according to the three guidelines also shows this trend (Figure 2b). The 2016 guidelines led to more bad tumours being classified as good, while the number of ugly tumours was not affected. However, in the 2020 guidelines, the criteria for ugly tumours changed, resulting in more than half of these tumours being classified as bad. Ultimately, according to the 2020 guidelines, half of all rectal tumours are early/good and only 15% locally advanced/ugly.

### 3.4. Selection to Different Treatments

The three main treatments were immediate surgery, scRT with surgery (immediate or delayed) and CRT/scRT + CT with delayed surgery. Overall, 475 (41%) patients were operated on directly and 641 (56%) patients were pre-treated. Of the remaining patients, 35 (3%), were handled differently (Figure 1). Among the pre-treated patients, 355 (55%) had scRT and 275 (43%) had CRT/scRT + CT (2% had long-course RT, *n* = 5, or CT, *n* = 5). These proportions varied between the regions and through the years. The main results are presented in Table 2 in relation to cTN-stage and risk group. Organ preservation was an option. It was, with rare exceptions, only practised if the tumour responded well. If there was an upfront intention, CRT was recommended in fit patients.

Overall, adherence to the guidelines was high. Most patients with an early/good tumour were operated on directly; 22% with a local procedure and 78% with an anterior resection, rectal excision, or a Hartmann’s procedure. Most patients with an intermediate/locally advanced/bad–ugly tumour received scRT or CRT/scRT + CT (Table 2). More patients from Uppsala were pre-treated (63% vs. 54%, *p* < 0.001). An increase in the proportion operated directly was seen in both regions with time (Figure 3a). Over the years, pre-treated intermediate/bad tumours received CRT/scRT + CT more often than scRT in both regions (*p* < 0.001, Figure 3b,c). The use of scRT alone decreased in locally advanced/ugly tumours. Consequently, the recommended treatment, CRT/scRT + CT, increased.

In univariable and multivariable regression analyses, tumour level and all relevant MRI characteristics were significantly associated with treatment decision (Table 3A). Separate analyses were performed for cT3 cases, revealing that MRF+ was important, as was the cT3-substage (explored for cT3ab vs. cd based on the division in the 2016 and 2020 guidelines but not in those from 2008). Year of diagnosis (2010–2014, 2015–2018 and 2019–2020 corresponding to when the guidelines were practised) was also significant, with less patients being pre-treated during the two recent periods. Age and region were associated with treatment choice in the univariable but not in the multivariable analyses. All MRI characteristics were also independently important for whether the patients were pre-treated with scRT or CRT/scRT + CT (Table 3B). EMVI was statistically significant on its own, but not in the multivariable analyses, indicating that it is closely related to other factors. In addition to these MRI characteristics, the year of diagnosis was independently important. Furthermore, region was important with more patients irradiated in Uppsala than in Dalarna. Older patients were more likely to receive scRT than CRT/scRT + CT and sex was not relevant in any analysis.

### 3.5. Correlations between Clinical and Pathological Stage

Since pre-treatment with a delay may result in down-staging, correlations between clinical and pathological stage were evaluated only in patients who were operated upfront or directly after scRT (within 21 days from the first fraction), i.e., chiefly in patients having early/good or intermediate/bad tumours. Congruence was fair for T-stage (accuracy 59%, K = 0.37, *p* < 0.001), slightly better in Uppsala than in Dalarna, and has essentially stayed the same over time (Appendix A). The congruence for cT3-substages was worse (accuracy 22%, K = 0.05, *p* = 0.07). For 30 MRF+ cT3-tumours operated on directly or usually after scRT with immediate surgery, no patient had a circumferential resection margin of < 1 mm (CRM+). Furthermore, the congruence for N-stage was poor (accuracy 54%, K = 0.16, *p* < 0.001), worse in Uppsala, but there seems to have been a slight improvement over time (Appendix A). Since these results may inaccurately disfavour the MR evaluations, an evaluation of pre-treated tumours with a delay was also made (see Appendix A for patient characteristics and main results).

## 4. Discussion

Tumour level, stage, other MRI characteristics and the risk grouping based upon these parameters were most relevant for the decision to operate on a rectal cancer patient directly or to initially provide non-surgical treatment in a large population-based patient sample. Age was of limited relevance whether to operate directly or not, although it influenced the type of therapy provided, and, in contrast to other studies [43,44], sex was not important. Thus, males were not pre-treated more often than females.

During the 11-year period, three national guidelines influenced the treatment decisions. There is a tradition in Sweden to follow medical recommendations provided by national experts, although this has not been formally documented. In general, adherence to the guidelines was at a high level, as most patients were treated according to the recommendations based upon the “good–bad–ugly” concept, but deviations were seen. Patient-related factors such as previous radiation, co-morbidities and personal wishes are relevant and can explain why a few (about 7%, slightly more in the region without a radiotherapy department) patients judged to have locally advanced/ugly tumours did not receive pre-treatment, while as much as one third of the intermediate/bad tumours did not receive pre-treatment, showing a larger tendency to deviate from the recommendations. The deviation from the guidelines for the early/good tumours was significant as about 18% received pre-treatment. A wish to avoid surgery, i.e., organ preservation, was initially not prominent, but increased with time. Rather, a wish to diminish the use of RT was important, but this ambition was, at least partly, counterbalanced by the Stockholm III trial closing patient entry in January 2013 [11]. The other major trial, the RAPIDO trial [41], did not impact on the proportion irradiated since only locally advanced/ugly tumours were included. It only affected the type of pre-treatment given (conventional CRT or experimental scRT + CT).

The intention to irradiate fewer patients was already present in 2010; however, it was not stressed in the 2008 guidelines, but expressed in the 2016 and, particularly, in the 2020 guidelines. This writing had a considerable influence on the proportion irradiated, meaning that even if there is an attempt to operate differently, it will not have much impact until changes have been made in the written guidelines. It was estimated that irradiation could be reduced by 5–10% applying the 2016 version rather than the 2008 version [19], which is in line with the results seen here. Prior to the 2020 revision, a similar estimate showed that between 10–15% fewer patients should be irradiated; a marked drop was also seen (see Figure 3). The proportion irradiated in these two (out of 21) regions in Sweden is similar to that in the whole of Sweden, revealing a slight decrease during the past decade [45]. The efforts to decrease irradiation was similar but expressed earlier and in a more pronounced way in Dalarna than in Uppsala, as the results disclose; the early/good and the locally advanced/ugly groups were similarly handled but in the intermediate/bad group, more patients were operated on directly in Dalarna than in Uppsala. The presence of a RT department in Uppsala may then be relevant, but the efforts of Swedish health care is to provide similar care to all citizens irrespective of where they live. This has not always been the case in colorectal cancer [43,44,46]. Patients from Dalarna have to travel at least 60 km to the irradiation facility and some need to travel up to 300 km, whereas in Uppsala, the maximum travel distance is 100 km.

Since any node positivity meant pre-treatment in the 2008 guidelines, the increase in the number of cases judged as cN+ until about 2014 meant that more patients were irradiated, counterbalancing the efforts to irradiate fewer patients, and more so in Uppsala than in Dalarna. In the 2016 guidelines, the stricter criteria for the lymph node evaluation were defined, possibly explaining the decrease in cN-positivity, particularly cN2, and not all cN1 should be irradiated. In the 2020 version of the guidelines, the relevance of the criteria was further stressed. A similar modification of the guidelines was made in the Netherlands in 2014, resulting in a decrease in the number of low-risk resectable rectal cancers irradiated and an increased specificity of nodal staging [47].

It is not possible to determine if the patients in Uppsala had more advanced tumours (more often cN+/cN2 disease, MRF+, and EMVI+) or if the radiological evaluation differed between the regions. It may be difficult to avoid differences in interpretation, but the teams of radiologists, having a focus on lower abdominal diseases, have participated in the same educational activities [35,36]. When images from patients from Dalarna to be irradiated were re-evaluated in Uppsala, the original evaluation was infrequently changed (results not shown). Colorectal cancer increases in the younger population, and they have a more aggressive disease [48,49]. These differences can explain why more patients received pre-treatment in Uppsala, having a younger population than in Dalarna. The patients in Uppsala also had more advanced tumours (more cN+, MRF+, EMVI+) that may support this notion. Tumour level and the proportion with metastatic disease were, however, similar between the regions. A limitation of the study is that tumour deposits on MRI were not evaluated, which has recently been advocated as important in the clinical staging [50]. Similarly, lateral node positivity was not frequently reported. Practically all tumours showing lateral node involvement are located below the peritoneal reflection and are considered as cT3/4, thus requiring pre-treatment anyhow. Both have been included in the 2020 guidelines and are presently relevant for risk grouping and treatment choice. Moreover, other factors that are not included in the guidelines, such as mucinous tumour, are also reported by radiologists and may have influenced the treatment decision [51,52].

The congruence of clinical and pathological N-stage was poor, with a tendency for over-staging. This was true for the early/good tumours; the overall accuracy of the MR evaluation is probably better for more advanced tumours [53,54]. Nodal stage is important in the treatment decision, but when the accuracy is so poor [55], it is questionable to consider it. The importance of N positivity in the surgical specimen for the risk of local recurrence/distant metastasis has been known for decades [56,57]. Nodal stage is also an important factor in the neo-adjuvant rectal score (NAR), providing prognostic information after neoadjuvant treatments [58]. Consequently, the evaluation of N-stage needs to be improved [59]. However, better criteria for N positivity would not mean much unless they can distinguish if the node is benign or malignant [31].

A strength is that this study is based on a large, unselected population. Missing data in the SCRCR was limited, and any information missing was retrieved from patient record files, resulting in close to 100% coverage. Due to the heterogeneity of the cohort and the division into several groups, too few patients may be included in some subgroups to draw firm conclusions, especially when comparing the different time periods, with the last one being only two years (2019–2020) and hence half the number of patients compared with the two other periods. Since our aim was to describe what determines treatment in clinical routine by “specially trained teams”, no independent review of the MR images was performed.

## 5. Conclusions

MRI characteristics, chiefly cTN-stage, MRF- and EMVI positivity, tumour level and treatment guidelines were important in determining pre-treatment in primary rectal cancer. Adherence to the guidelines was high but some variability due to age and inclusion in ongoing clinical trials was seen. Changes in the use of RT/CRT over time were closely related to changes in the guidelines. One of the guideline changes was prompted by a “difficult-to-explain” increase in cN positivity, probably responsible for an increased use of RT/CRT during the first years of the study period. Furthermore, the accuracy of the MRI evaluations was not high, especially not for N-staging, and needs improvement. Education in rectal cancer MRI staging and multidisciplinary work does not guarantee high accuracy in clinical routines. Fewer pre-treated patients in the intermediate/bad group in one region indicate that the handling differed. This could be the result of a greater strife to diminish irradiation, but a longer distance to the radiotherapy department, and thus, an inequality in care cannot be ruled out.

## Figures and Tables

**Figure 1 cancers-15-01154-f001:**
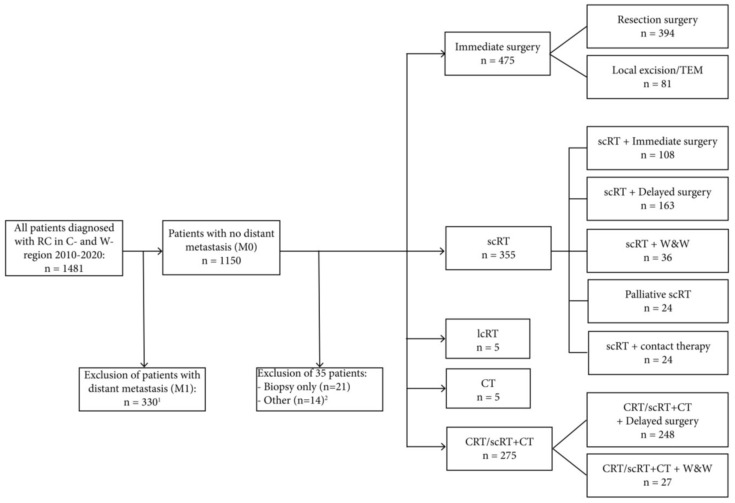
Flow chart of patient selection. ^1^ One patient had a missing cTNM status and was excluded. ^2^ Patients that were too ill to be treated (*n* = 4), died before initiating treatment (*n* = 2), carcinoma in situ (*n* = 2), another metastatic tumour type simultaneously (*n* = 2), sigmoidal cancer (*n* = 1), missing information about treatment (*n* = 1), M1 according to pathology report (*n* = 1), or no tumour could be seen on MRI (*n* = 1). Abbreviations: Rectal cancer, RC; short-course radiotherapy, scRT; long-course radiotherapy, lcRT; chemotherapy, CT; chemoradiotherapy, CRT; transanal endoscopic microsurgery, TEM; watch-and-wait strategy, W&W.

**Figure 2 cancers-15-01154-f002:**
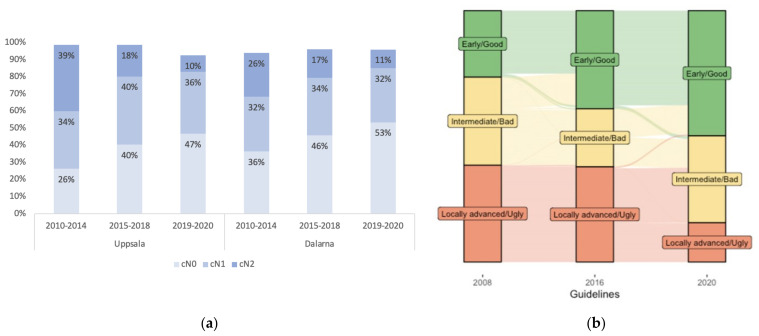
(**a**) Distribution of cN-stage for the time periods during which different treatment guidelines were used (2008 guidelines in 2010–2014, 2016 guidelines in 2015–2018 and 2020 guidelines in 2019–2020) in Uppsala and Dalarna regions. The columns do not add up to 100% since nodal status was not detailed in all patients. The changes in cN positivity for each year is shown in Appendix A; (**b**) Distribution of risk groups in an alluvial diagram after re-classifying all tumours (that had all necessary data available, *n* = 1038) according to the different guidelines. With the 2008 guidelines, there was an even distribution of the three risk groups; 26% of the tumours were classified as early/good, 36% intermediate/bad and 38% locally advanced/ugly. With the 2016 guidelines, the number of bad tumours decreased while the number of ugly tumours stayed virtually the same; 38% good, 24% bad and 37% ugly. Major changes in the 2020 guidelines meant that even more tumours should be considered good, and more than half of the ugly tumours should now be considered bad; 50% good, 35% bad and 15% ugly.

**Figure 3 cancers-15-01154-f003:**
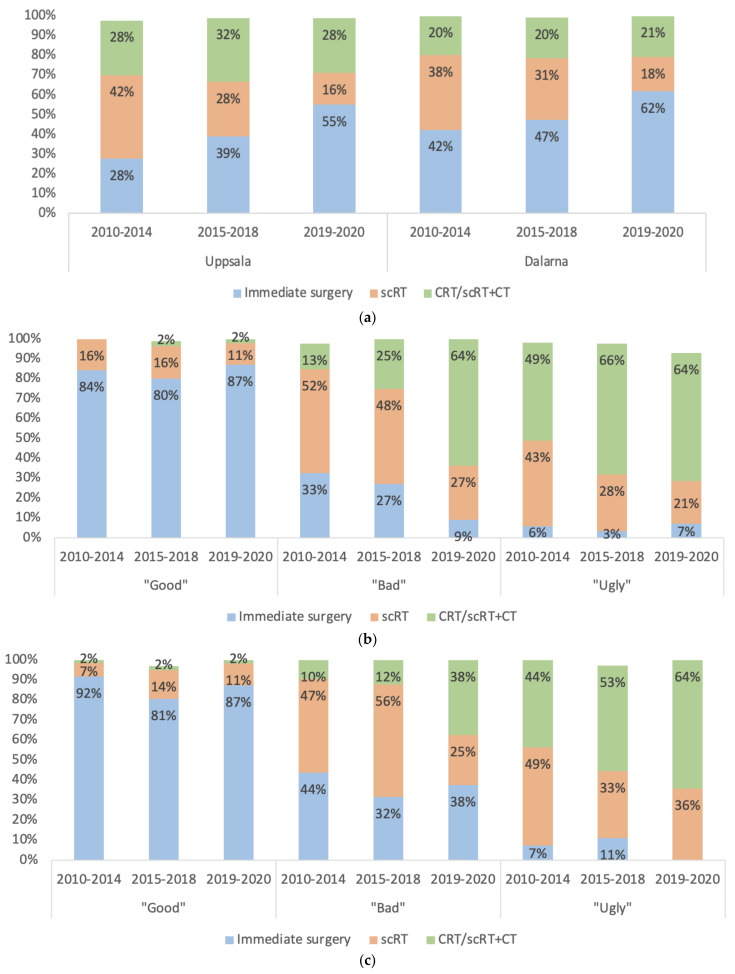
Distribution of treatments (immediate surgery, short-course radiotherapy; scRT, chemoradiotherapy/scRT + chemotherapy; CRT/scRT + CT) in (**a**) both regions and also in each risk group (Good, Bad, Ugly) in (**b**) Uppsala and (**c**) Dalarna region for the time periods during which different treatment guidelines were used (2008 guidelines in 2010–2014, 2016 guidelines in 2015–2018 and 2020 guidelines in 2019–2020). The columns do not add up to 100% since some patients were treated with other alternatives than the three dominating ones.

**Table 1 cancers-15-01154-t001:** Characteristics at baseline and chosen initial treatment for patients without distant metastasis (M0).

**Characteristics**	**Uppsala (C-Region)** ***n* = 542**	**Dalarna (W-Region)** ***n* = 608**	** *p* ** **-Value**
Age, median	70 (32–96)	73 ^1^ (26–96)	<0.001
Men	314 (58)	357 (59)	0.788
cT-stage ^2^			0.228
T1	40 (7)	49 (8)	
T2	104 (19)	131 (22)	
T3	275 (51)	291 (48)	
a	68 (25)	49 (17)	
b	111 (40)	118 (41)	
c	74 (27)	76 (26)	
d	15 (6)	17 (6)	
T4	109 (20)	112 (18)	
a	20 (18)	43 (38)	
b	83 (76)	67 (60)	
cN-stage			0.003
N0	193 (36)	260 (43)	
N1	199 (37)	198 (33)	
N2	137 (25)	121 (20)	
MRF+ (in cT3 tumours ^3^)	107 (39)	81 (28)	0.005
EMVI+	147 (28)	134 (23)	0.045
Tumour level			0.205
Low (0–4 cm)	119 (22)	150 (25)	
Mid (5–9 cm)	191 (35)	219 (36)	
High (10–15 cm)	230 (42)	238 (39)	
Treatment			<0.001
Immediate surgery	196 (39)	278 (46)	
scRT	170 (31)	186 (31)	
CRT/scRT + CT	157 (31)	118 (21)	
Risk group ^4^			0.007
Early/good	169 (31)	221 (36)	
Intermediate/bad	151 (28)	172 (28)	
Locally advanced/ugly	210 (39)	187 (31)	

^1^ The population in Dalarna is older than in Uppsala and the rest of Sweden with, e.g., 25% of the population above 65 years compared to 20%. ^2^ Depending upon the depth of infiltration, cT3 was subdivided into a–d (<1 mm, 1–5 mm, 5–15 mm and >15 mm) and cT4 into involvement of peritoneum only (a) or other organs (b). ^3^ MRF+ indicates that the distance is either <1 mm (threatened) or 0 mm (involved); it was evaluated only in cT3 tumours, thus not in cT2 or cT4a tumours. cN1 means less than four while cN2 means four or more mesorectal lymph nodes are considered involved. Lateral nodes were considered positive if the largest diameter was at least 10 mm (LN+); however, it was not possible to register the presence of LN until after 2017 and the presence of these was not frequently commented upon until after about 2013, nor was it possible to register the presence of tumour deposits (TD) until after 2017 and comments about whether they were present or not were not common until the two most recent years. Thus, LN or TD positivity is not included in the table. ^4^ Criteria for risk groups described in Appendix A. Abbreviations: Clinical tumour stage, cT-stage; clinical nodal stage, cN-stage; Mesorectal fascia, MRF; extramural vascular invasion, EMVI; short-course radiotherapy, scRT; chemoradiotherapy, CRT; chemotherapy, CT.

**Table 2 cancers-15-01154-t002:** Distribution of age, different MRI characteristics, tumour level and risk groups in patients without distant metastases (M0) that had immediate surgery, short-course radiotherapy (scRT) and chemoradiotherapy (CRT)/scRT + chemotherapy (CT).

Treatments	Immediate Surgery *n* = 475	scRT ^1^ *n* = 355	CRT/scRT + CT ^1^ *n* = 275	*p*-Value
**Age**				<0.001
<65 years	118 (25)	67 (19)	122 (44)	
65–79 year	255 (54)	158 (45)	149 (54)	
≥80 years	102 (21)	130 (37)	4 (1)	
**Men**	276 (58)	208 (58)	162 (59)	0.966
**cT-stage**				<0.001
T1	80 (17)	4 (1)	1 (0.4)	
T2	168 (35)	54 (15)	9 (3)	
T3	194 (41)	225 (63)	133 (48)	
a	63 (32)	47 (21)	10 (8)	
b	96 (50)	103 (46)	35 (26)	
c	29 (15)	57 (25)	68 (51)	
d	1 (0.5)	11 (5)	20 (15)	
T4	9 (2)	69 (19)	131 (48)	
a	7 (78)	15 (22)	39 (30)	
b	2 (22)	53 (77)	92 (70)	
**cN-stage**				<0.001
N0	312 (66)	98 (28)	27 (10)	
N1	121 (26)	164 (46)	101 (37)	
N2	16 (3)	91 (26)	147 (53)	
**MRF+ (in cT3-tumours)**	15 (8)	77 (34)	89 (67)	<0.001
**EMVI+**	32 (7)	104 (29)	140 (51)	<0.001
**Tumour level**				<0.001
Low (0–4 cm)	62 (13)	111 (31)	84 (31)	
Mid (5–9 cm)	150 (32)	150 (42)	93 (34)	
High (10–15 cm)	261 (55)	94 (26)	98 (36)	
**Risk group**				<0.001
Early/Good	318 (67)	48 (14) ^2^	6 (2) ^2^	
Intermediate/Bad	110 (23)	154 (43)	61 (22)	
Locally advanced/Ugly	24 (5) ^3^	148 (42)	207 (75)	

^1^ Patients that had a watch-and-wait strategy (W&W) are included; scRT n = 36, CRT/scRT + CT n = 28. In the scRT-group, 34 patients were enrolled in the Stockholm II trial. In the CRT/scRT-CT group, 109 patients were in the RAPIDO trial and 89 patients in the LARCT-US trial (see Appendix A for details). ^2^ Reasons for an early/good tumour to receive RT/CRT were a wish for organ preservation (*n* = 12, six had CRT/scRT + CT and the rest scRT), palliation in non-operable patients (*n* = 5, all had scRT), contact therapy (*n* = 8), very low tumour and/or case for an abdomino-perineal resection (*n* = 7), co-morbidity (*n* = 7), tumour stage cT3bN1 (*n* = 2), suspected metastasis (M1, *n* = 2), and, prior to February 2013, inclusion in the Stockholm III trial (*n* = 3). One patient wished for pre-treatment, one patient had a myocardial infarction prior to diagnosis, and one patient received scRT while waiting too long for surgery. The reason was not recorded in 5 patients. ^3^ Reasons not to irradiate an ugly tumour were prior RT (*n* = 2), severe comorbidity (*n* = 3), being considered to have low risk of local recurrence (*n* = 4), and a very high, rectosigmoid location in a tumour considered to be easily resected (*n* = 9). Information was missing in six patients. Abbreviations: Clinical tumour stage, cT-stage; clinical nodal stage, cN-stage; mesorectal fascia, MRF; extramural vascular invasion, EMVI.

**Table 3 cancers-15-01154-t003:** (**A**) Univariable and multivariable logistic regression for patients that had preoperative treatment (short-course radiotherapy (scRT) or chemoradiotherapy/scRT + chemotherapy) (*n* = 630) compared to patients that had no preoperative treatment (*n* = 475). (**B**) Univariable and multivariable logistic regression for patients that had chemoradiotherapy (CRT) or short-course radiotherapy(scRT) + chemotherapy (CT) (*n* = 275) compared to patients that had only scRT (*n* = 355).

**(A)**						
**Pre-Treatment or Not**	**Univariable Logistic Regression**			**Multivariable Logistic Regression**		
	**OR**	**95% CI**	** *p* **	**OR**	**95% CI**	** *p* **
**Sex (Male)**	0.968	0.761–1.233	0.795	-	-	-
**Age**	0.903	0.761–1.071	0.242	-	-	-
**Region (Dalarna)**	0.655	0.515–0.834	<0.001	0.606	0.421–0.872	0.007
**Year of diagnosis**	0.625	0.527–0.741	<0.001	0.621	0.483–0.798	<0.001
**Tumour level**	0.478	0.405–0.564	<0.001	0.211	0.159–0.281	<0.001
**cT-stage**	5.961	4.702–7.557	<0.001	4.550	3.327–6.225	<0.001
**cT3-substage ***	4.240	2.722–6.605	<0.001	-	-	-
**cN-stage**	5.766	4.615–7.204	<0.001	3.720	2.784–4.970	<0.001
**MRF+ (cT3 tumours) ***	9.683	5.481–17.109	<0.001	-	-	-
**EMVI+**	7.930	5.338–11.781	<0.001	3.128	1.897–5.158	<0.001
**(B)**						
**CRT/scRT + CT or scRT alone**	**Univariable Logistic Regression**			**Multivariable Logistic Regression**		
	**OR**	**95% CI**	** *p* **	**OR**	**95% CI**	** *p* **
**Sex (Male)**	0.992	0.720–1.365	0.959	-	-	-
**Age**	0.247	0.187–0.326	<0.001	0.179	0.125–0.255	<0.001
**Region (Dalarna)**	0.687	0.500–0.943	0.020	0.638	0.426–0.957	0.030
**Year of diagnosis**	1.539	1.219–1.942	<0.001	2.182	1.584–3.005	<0.001
**Tumour level**	1.175	0.960–1.437	0.117	-	-	-
**cT-stage**	3.175	2.373–4.247	<0.001	3.800	2.585–5.587	<0.001
**cT3-substage ***	4.314	2.725–6.830	<0.001	-	-	-
**cN-stage**	2.421	1.910–3.069	<0.001	2.226	1.612–3.073	<0.001
**MRF+ (cT3 tumours) ***	3.678	2.331–5.802	<0.001	-	-	-
**EMVI+**	2.347	1.683–3.272	<0.001	1.376	0.892–2.123	0.149

Age grouped by <65, 65–79, 80+ years. Year of diagnosis grouped according to which national treatment guideline was used; 2010–2014 (2008 guidelines), 2015–2018 (2016 guidelines), 2019–2020 (2020 guidelines). Tumour level grouped by Low (0–4 cm), Mid (5–9 cm), High (10–15 cm). cT3-substage grouped in cT3ab and cT3cd. * cT3-substage and MRF+ excluded from the multivariate analysis as all tumours in these categories could only be cT3 (see Table 1). Abbreviations: Clinical tumour stage, cT-stage; clinical nodal stage, cN-stage; mesorectal fascia, MRF; extramural vascular invasion, EMVI.

## Data Availability

The data presented in this study are available upon reasonable request from the corresponding author.

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
