# Peer review of "Determinants of Pre-Surgical Treatment in Primary Rectal Cancer: A Population-Based Study"

_cancers, 2023, doi:10.3390/cancers15041154_

Round 1

Reviewer 1 Report

Intersting small differences between to regions.

Author Response

Thank you for the comment.

Reviewer 2 Report

I would recommend a a minor check of the abstract to make more fluent

Author Response

Thank you for the suggestion. The abstract has been checked. Due to a word-limit only smaller changes were possible to make.

Reviewer 3 Report

This paper is very significant and can be acceptable.

Author Response

Thank you.

Reviewer 4 Report

Thank you for the possibility to review the manuscript titled: “Determinants of pre-surgical treatment in primary rectal cancer: A population-based study”. The study presents an overview of rectal cancer diagnosis and management from tho regions of Sweden. There are several improvements that should be taken into account:

-Please review the language of the manuscript.

-The study analyses two regions of Sweden, however the discussion section mostly pays attention to Swedish guidelines for rectal cancer. There should be more information comparing the regions.

-It is not completely understood why the two regions were compared and wht is the purpose. Please pay more attention to the reason of comparison

-The adherence to guidelines should be underlined. It is hard to understand based on the results how did this conclusion appear.

-Please explain how was kappa calculated. Was this interobserver agreement between two regions? How were the radiologists assessed then?

Please take into account the recommendations in the spirit of improving the quality of the submission.

Author Response

Comment 1: Please review the language of the manuscript.

Response: The language has now been reviewed by the authors and externally by a native English-speaking person.

Comment 2: The study analyses two regions of Sweden, however the discussion section mostly pays attention to Swedish guidelines for rectal cancer. There should be more information comparing the regions.

Response: This has been addressed by adding the following text in the manuscript: “…fewer patients, more so in Uppsala than in Dalarna” (page 10, line 447), and “The patients in Uppsala also had more advanced tumours (more cN+, MRF+, EMVI+) that could support this notion” (page 11, line 467-468).

Comment 3: It is not completely understood why the two regions were compared and what is the purpose. Please pay more attention to the reason of comparison

Response: The two regions were compared with the purpose to evaluate whether the care is equal, i.e., whether unmotivated differences exist. The following sentence has been added in the introduction, page 2 line 65-66: “Two adjacent Swedish regions were compared for the evaluation of equal care, being an important aspect of Swedish health care.”

Comment 4: The adherence to guidelines should be underlined. It is hard to understand based on the results how did this conclusion appear.

Response: The following words have been added in the text to make this clearer: “Overall, adherence to the guidelines was high.” (page 6, line 226). “…as most patients were treated according to the recommendations based upon the “good-bad-ugly” concept…” (page 10, line 411-412).

Comment 5: Please explain how was kappa calculated. Was this interobserver agreement between two regions? How were the radiologists assessed then?

Response: Kappa was calculated for the agreement between clinical and pathological T- and N-stage. The information used was directly from the SCRCR. It was only done in patients that had been operated directly (with or without 5x5 Gy prior) since pre-treatment with delay may lead to downstaging. Consequently, in the cases investigated the pathological stage should be the “true” stage. The kappa gives then an indication of how correct the clinical staging is. This was done for all cases and then separately for each region.

Round 2

Reviewer 4 Report

The authors have made all of the necessary corrections.